# Comparison of longitudinal PSQI and actigraphy-assessed sleep from pregnancy to postpartum

Ryan L. Brown[1], Chloe M. Beverly Hery[2], Aric A. Prather[3], Lisa M. Christian[4,5]*

1 Department of Human Development and Family Sciences, Texas Tech University, Lubbock, Texas, United States of America, 2 Comprehensive Cancer Center, The Ohio State University, Columbus, Ohio, United States of America, 3 Department of Psychiatry and Behavioral Sciences, University of California, San Francisco, California, United States of America, 4 Department of Psychiatry & Behavioral Health, The Ohio State University Wexner Medical Center, Columbus, Ohio, United States of America, 5 Institute of Behavioral Medicine Research, The Ohio State University Wexner Medical Center, Columbus, Ohio, United States of America

* Lisa.Christian@osumc.edu

## Abstract

Despite the importance of sleep for perinatal health, there is limited research examining whether different measurement modalities may yield inconsistent data from pregnancy through postpartum. We aimed to: 1) describe how sleep patterns change across pregnancy and postpartum using self-report PSQI and actigraphy measures and 2) determine the level of correspondence between these two measurement modalities. Pregnant women from the Stress and Health in Pregnancy and Postpartum (SHIPP) study completed visits during the 3rd trimester, 4–6 weeks postpartum, and 4 months, 8 months, and 12 months postpartum. At each study visit, participants completed questionnaires and wore wrist-actigraphy (Actiwatch 2) for one week prior to each visit. Self-reported global sleep quality was measured with the Pittsburgh Sleep Quality Index (PSQI). Actigraphy and self-reported PSQI sleep characteristics were summarized at each of the five study visits. Using generalized linear mixed modeling, we examined if there were differences by sleep measurement (actigraphy vs. PSQI) for the overlapping sleep outcomes: total sleep time, time in bed, sleep latency, and sleep efficiency. Participants ($n = 74$; 28.9 years ± 4.6) were mostly white (73%), non-Hispanic (96%), married (78.5%) and over 60% had at least one child previously. Average PSQI global score was > 5 (cutoff for poor sleep) at each study visit. Total sleep time, sleep efficiency, and sleep latency measurements were significantly different between self-report and actigraphy throughout pregnancy and postpartum. Actigraphy-assessed sleep may reflect longer total sleep times, shorter sleep latency, and greater sleep efficiency compared to self-reported sleep among pregnant and postpartum women. This may be due to measurement error in actigraphy or recall bias when completing self-reported sleep measures. These factors should be taken into consideration both at the time of study design and when comparing results from different studies to facilitate the highest quality research and clinical decision-making in this population.

**Data availability statement:** All relevant data are within the manuscript and its Supporting Information files.

**Funding:** This study was supported by the Eunice Kennedy Shriver National Institute of Child Health and Human Development (NICHD) [grant numbers HD061644 and HD067670 to L.M.C.]. The project described was also supported by Award Number UM1TR004548 from the National Center for Advancing Translational Sciences. The content is solely the responsibility of the authors and does not necessarily represent the official views of the National Center for Advancing Translational Sciences or the National Institutes of Health. The funders had no role in study design, data collection and analysis, decision to publish, or preparation of the manuscript.

**Competing interests:** The authors have declared that no competing interests exist.

## Introduction

Pregnant and postpartum women commonly experience various sleep issues [1]. Sleep issues that arise during pregnancy tend to be pain-related insomnia, positional discomfort, leg cramps, and frequent urination [2,3], whereas sleep issues during postpartum are more often fragmented or interrupted sleep and shorter sleep durations due to infant sleep schedules [4,5]. Poor maternal sleep can impact the mother's mental health and functioning as well as increase the risk of adverse pregnancy outcomes, such as intrauterine growth restriction and preterm birth [6–11]. Accordingly, assessing sleep is a focus of many studies on perinatal mental health.

However, the measurement modality used to assess sleep characteristics may be affected differently across pregnancy and postpartum based on stage of pregnancy and the needs of one's infant. While pregnant, women report more snoring late in their pregnancy and increases in frequency of restless less syndrome and daytime sleepiness [12]. Hormonal changes during pregnancy can also affect sleep, including altering sleep architecture [13]. In contrast, postpartum sleep disturbances are characterized by nighttime awakenings from infant sleep and feeding schedules, resulting in poor sleep quality and shortened sleep duration, most often for mothers [4,5,14].

Despite the substantial literature on sleep during pregnancy and postpartum, most studies in this area are cross-sectional with few studies quantifying changes in sleep over time from pregnancy through postpartum [15,16]. Poor sleep quality generally increases throughout pregnancy, peaks during the first few months postpartum, and tends to remain elevated for the first six months to one year [3,17,18]. Yet, repeated measures using self-report and objective measures of sleep changes, such as actigraphy, across the perinatal period are lacking, with only a few exceptions [19–22]. There are clear benefits and challenges to measuring sleep subjectively in new parents. Self-reported sleep, including questionnaires or sleep diaries, is an inexpensive and low burden option; however, self-report measures are subject to recall bias and depend on the participant to accurately and fully complete all portions of the sleep assessment [23]. While polysomnography is considered the gold standard of sleep research, it can be costly and difficult to schedule in-laboratory visits for new parents. Therefore, more research uses wrist-worn devices to collect actigraphy data to overcome these barriers and provide less obtrusive, objective sleep measures [23,24].

When considering actigraphy-assessed sleep throughout the perinatal period, there are several limitations to the available literature such as small samples and a lack of postpartum assessments. Six studies in the United States have examined changes in parental sleep from pregnancy to postpartum using actigraphy [25–30], of which four had small sample sizes of ≤30 participants [27–30]. Additionally, none of these studies have examined sleep after 16 weeks postpartum, when many sleep complaints do not begin to occur until the third trimester (after 28 weeks) and few report sleep metrics beyond total sleep time and sleep efficiency. In the current study, we report actigraphy-assessed sleep from third trimester to one year postpartum to address this gap in postpartum measurement.

A limited number of studies have examined differences between self-reported sleep measures (e.g., sleep diary or the Pittsburgh Sleep Quality Index (PSQI)) and

actigraphy collected sleep measures concurrently among pregnant and postpartum women [21,31–33]. Of these, only two have compared the self-report measure to the actigraphy measures of sleep during pregnancy [21,31]. In these studies, there were discrepancies identified based on measurement modality for sleep duration, wake after sleep onset, and sleep efficiency during pregnancy [31]. Over 60% of participants self-reported sleep duration estimates that differed by over an hour compared to actigraphy [21], actigraphy-assessed wake after sleep onset was 3.5 times greater than self-reported [31], and self-reported sleep efficiency was much higher than actigraphy-assessed [31]. More studies are needed to determine the robustness of these results and extend the question into the postpartum period. Thus, the objectives of this study were to 1) describe how sleep patterns change across pregnancy and postpartum using both PSQI self-reported sleep measures and actigraphy measures and 2) determine the level of correspondence between these two complementary measurement modalities.

## Materials and methods

### Study design

The Stress and Health in Pregnancy and Postpartum (SHIPP) study included pregnant women from the central Ohio area. Briefly, this study was designed to examine associations between psychosocial factors, mood, health behaviors, neuroendocrine and immune function from the pregnancy through postpartum. Study visits were conducted during the 3rd trimester, 4–6 weeks postpartum, and 4 months, 8 months, and 12 months postpartum. Participants completed questionnaires and psychosocial interviews at each visit and wrist-actigraphy was collected data for one week prior to each in-person study visit. Recruitment occurred from December 8, 2016, to April 9, 2018. The current analyses focus on self-reported sleep as well as actigraphy-based sleep data. The study methods and results follow best practice recommendations and reporting guidelines for Strengthening the Reporting of Observational Studies in Epidemiology (STROBE).

### Participants

Women were eligible for this study if they were between the ages of 18–40 years old and were between 29–34 weeks gestation. Women were excluded if they had multifetal gestation, fetal anomaly, chronic health problems with a clear immunological or endocrinological component (e.g., cancers, recent surgeries, strokes, or conditions that require regular use of anti-inflammatories), worked night shifts, or reported alcohol use or illicit drug use (other than marijuana). In addition, women who were underweight (pre-pregnancy body mass index, BMI, $< 18.5 \, \text{kg/m}^2$) or had a BMI $> 50 \, \text{kg/m}^2$ were excluded. Women reporting recent acute illness (e.g., cold) or vaccinations in the past 7 days were rescheduled for a later visit. Written informed consent and Health Insurance Portability and Accountability Act (HIPAA) authorizations were obtained from all participants and each received modest compensation. The study was approved by the OSU Biomedical Institutional Review Board.

A total of 83 women were enrolled in this study with 74 providing actigraphy data at baseline and at least one other time point. For each assessment window, data were only used if ≥ 3 days of valid actigraphy data were collected, as this has been deemed a minimum criterion for reliable and valid data [34]. Across all five study visits, there were only 23 cases (~8% of data points) of individuals with only three to four nights of actigraphy data recorded. Therefore, the analyzed sample included 59 participants at visit 1, 66 at visit 2, 60 at visit 3, 62 at visit 4, and 56 at visit 5.

### Demographic data

Age, race/ethnicity, marital status, education, income level, and parity, were self-reported. Pre-pregnancy body mass index (BMI) was calculated using self-reported pre-pregnancy weight and measured height at the first visit. Length of gestation at each visit was calculated from the visit date and the expected due date. Length of gestation at delivery was calculated from the date of delivery.

## Self-reported sleep

At every study visit, subjective sleep was measured using the Pittsburgh Sleep Quality Index (PSQI), an 18-item measure of sleep quality over the past four weeks [35]. A PSQI global score is calculated by summing all seven component responses and ranges from 0 to 21. A higher score indicates worse sleep quality and a global score of ≥ 5 is the standard cut-off for poor sleep [35]. The individual PSQI items for hours in bed, sleep duration, sleep latency, and a calculated percent sleep efficiency were used in this analysis.

## Actigraphy-measured sleep

Wrist actigraphy data were collected for 7 days prior to each study visit using the Actiwatch 2 (Philips Respironics, Murrysville, PA, USA). Participants were instructed to wear the actiwatch on their non-dominant wrist for 24 hours per day. Actigraphy data collected from the Actiwatch 2 were downloaded using the Actiware Sleep Scoring Program (v6.0.9, Philips-Respironics).

Sleep actigraphy data files for each participant were scored in Actiware using the validated manufacturer's standard algorithm at medium sensitivity [34]. Two independent scorers determined the validity of each recording based on standard sleep criteria, adapted from other actigraphy studies [36–38]. A recording could be considered invalid for two reasons: 1) an Actiwatch device malfunction indicated by false activity and 2) Actiwatch data was unable to be retrieved. Individual days within a recording could also be considered invalid if an Actiwatch error occurred, such as a failing battery indicated by false activity, or if the participant did not wear the watch for an appropriate amount of time per day (i.e., majority of day off-wrist, or an off-wrist period > 60 minutes within 10 minutes of the determined beginning or end of the sleep period). Two scorers reviewed the automated rest intervals from Actiware and manually adjusted the interval as needed based on evidence of increased or decreased activity levels, rapidly decreased light levels to suggest bedtimes, and rapidly increased light levels to suggest wake times. For example, decreasing activity levels refers to the scorers considering how the activity intensity decreases surrounding periods of suspected sleep within the context of the entire participant's actigraphy profile. Scorers visually reviewed the entire sleep recording before manually adjusting any sleep periods to consider the activity intensity level of the participant to ensure periods of low movement, but not sleep (e.g., reading in bed, watching TV) were not included. Sleep diary data was used to help suggest, but not confirm, sleep periods as needed.

Each file was reviewed by an unbiased 3rd scorer for interrater reliability. For any interval with bed and wake times that differed by ≥ 15 minutes between scorers, the 3rd scorer reviewed and corrected those intervals accordingly. Any disagreements in scoring differences were adjudicated by the primary study investigator (LMC). Resulting actigraphy variables calculated from the scored sleep intervals included: time in bed, total sleep time, wake after sleep onset, sleep efficiency, and sleep latency.

## Statistical analysis

Participant characteristics at baseline were described using means with standard deviations (SD) for continuous variables and frequencies with percentages for categorical variables. Actigraphy and self-reported sleep characteristics were summarized (means with SD) at each of the five study visits. Cross-sectional correlations (i.e., at each time point from late pregnancy to postpartum) reflect the correspondence between actigraphy and PSQI measures at each time point throughout the transition to parenthood.

For the primary analyses, we aimed to determine if there were significant differences in each overlapping sleep outcome measure based on the method of data collection and if these differences emerged at particular time points during pregnancy or postpartum. We used generalized linear mixed modeling (GLMM), a multilevel regression analytic technique, to fit each outcome variable (i.e., time in bed, total sleep time, sleep latency, sleep efficiency) across each model: first as a function of the main effect of method of data collection (self-report vs. actigraphy) and then based on the

interaction between time (visit number) and method of data collection (self-report vs. actigraphy). In post-hoc sensitivity analyses, we confirmed that the presented results were unchanged when restricting the sample to only those who had five or more valid nights of actigraphy.

Before testing each primary model, we used a model comparison approach to determine the functional form of the relationship for each outcome with respect to linearity and random effects (i.e., testing whether the slope between time and sleep efficiency varied randomly, with variation based on the individual). The −2 log likelihood was used as an index to determine whether the model with the random intercept for each individual fit the data better than the model without the random intercept. We examined the significance test associated with the likelihood ratios between the models with and without the relevant parameters to assess the relative quality between the models based on the inclusion of random slopes and a linear versus quadratic effect of time. All analyses were conducted in the R statistical computing environment (R Core Team, 2023). Multilevel analyses were performed using the package *lme4* [39] with model comparisons tested within the *nlme* [40] package, and *ggplot2* [41] and *ggeffects* [42] were used for visualization.

## Results

### Descriptive statistics

There were 74 women who provided actigraphy data for at least two time points. On average, participants were 28.9 years old with a pre-pregnancy BMI of 27.1 kg/m$^2$; a majority were white (75.7%), married (78.5%), had a family income over $75,000 (50.6%), and had at least a college degree (74.7%). Most women had a prior birth (60.7%) and were an average 30.8 weeks gestation at the start of the study (see Table 1). On average these participants had between 6.3 and 6.6 valid days of actigraphy.

### Time in bed

Actigraphic time in bed and self-reported time in bed were similar, ranging from 7.97–8.66 hours by actigraphy and 7.72–8.52 hours by self-report (see Table 2; see Fig 1A). Accordingly, actigraphic time in bed and self-reported time in bed were consistently correlated at each time point captured in this study, with the strength of the association varying from low to high: 3$^{rd}$ trimester ($r = .58$, $p < .001$), 4–6 weeks postpartum ($r = .37$, $p = .002$), 4 months postpartum ($r = .62$, $p < .001$), 8 months postpartum ($r = .49$, $p < .001$), and 12 months postpartum ($r = .37$, $p = .005$).

Overall, there was a significant decrease in time in bed across the study visits ($b = −0.16$, 95% CI [−0.23, −0.09], $p < .001$); there was not a statistically significant difference in time in bed based on the method of data collection ($b = −0.10$, 95% CI [−0.27, 0.07], $p = .25$) and no significant interaction between time and method ($b = 0.00$, $p = .96$; see Fig 2A).

### Total sleep time

Actigraphic total sleep time was consistently higher than self-reported total sleep time until 12 months postpartum, ranging from 6.78–7.27 hours by actigraphy and 5.86–6.83 by self-report (see Table 2; see Fig 1B). However, actigraphic and self-reported total sleep time were reliably correlated across each time point in this study, with the strength of the association varying from low to moderate: 3$^{rd}$ trimester ($r = .34$, $p = .008$), 4–6 weeks postpartum ($r = .46$, $p < .001$), 4 months postpartum ($r = .40$, $p = .001$), 8 months postpartum ($r = .27$, $p = .040$), and 12 months postpartum ($r = .48$, $p < .001$).

These data showed a significant linear ($b = −0.67$, 95% CI [−0.94, −0.39], $p < .001$) and quadratic ($b = 0.11$, 95% CI [0.06, 0.15], $p < .001$) effect of time for total sleep time. There was also a significant difference based on the method of data collection ($b = −0.56$, 95% CI [−0.71, −0.42], $p < .001$) such that the actigraphic measure of sleep estimated longer sleep times than self-report measures of sleep across the visits. There was also a significant interaction between time (linear) and method of data collection ($b = −0.59$, 95% CI [−1.12, −0.06], $p = .029$), as well as time (quadratic) and method

**Table 1. Demographic characteristics of participants at baseline.**

| Characteristic | All Participants (n = 74) |
|---|---|
| **Age,** mean (SD) | 28.9 (4.4) |
| **Pre-Pregnancy BMI,** mean (SD) | 27.1 (6.4) |
| **Race** | |
| White | 56 (75.7) |
| Black | 12 (19.0) |
| Asian | 2 (2.5) |
| More than one race | 4 (5.1) |
| **Ethnicity** | |
| Hispanic | 2 (2.7) |
| Non-Hispanic | 72 (97.3) |
| **Marital Status** | |
| Married | 60 (78.5) |
| In a relationship | 12 (16.5) |
| Single | 2 (5.0) |
| **Total Family Income** | |
| <$30,000 | 9 (15.2) |
| $30,000–74,999 | 25 (34.2) |
| ≥ $75,000 | 40 (50.6) |
| **Education** | |
| High School Graduate or less | 6 (10.1) |
| Some College | 9 (15.2) |
| College Degree (2 year or 4 year) | 25 (31.7) |
| Graduate School | 34 (43.0) |
| **Parity** | |
| 0 | 31 (39.3) |
| 1 child | 31 (43.0) |
| 2+children | 12 (17.7) |
| **Length of gestation at baseline, weeks (SD)** | 30.8 (2.0) |
| **Length of gestation at delivery, weeks (SD)** | 39.4 (1.1) |

of data collection (b = 0.13, 95% CI [0.04, 0.21], $p$ = .005); an analysis of simple slopes indicated that this difference was significant in 3rd trimester of pregnancy (b = −0.57, $p$ < .001), 4–6 weeks postpartum (b = −1.01, $p$ < .001), 4 months postpartum (b = −0.57, $p$ < .001), and 8 months postpartum (b = −0.63, $p$ < .001), but the two measures converged by 12 months postpartum (b = 0.06, $p$ = .71; see Fig 2B). In sum, self-report and actigraphic estimations of total sleep time may differ by 30–60 minutes from one's 3rd trimester of pregnancy to 8 months postpartum but estimates are similar by 12 months postpartum.

## Sleep efficiency

Sleep efficiency followed a similar pattern in both methods; however, actigraphy-measured sleep efficiency was consistently higher than self-report, ranging from 79.88–86.17% by actigraphy and 69.44–84.18% by self-report (see Table 2; see Fig 1C). Actigraphic and self-reported sleep efficiency were only reliably correlated at 12 months postpartum ($r$ = .35, $p$ = .009), but not at any other study visit: 3rd trimester ($r$ = .06, $p$ = .66), 4–6 weeks postpartum ($r$ = .21, $p$ = .09), 4 months postpartum ($r$ = .20, $p$ = .14), nor 8 months postpartum ($r$ = .18, $p$ = .18).

**Table 2. Mean (SD) of actigraphy and PSQI measures from 3ʳᵈ trimester to 12 months postpartum (PP).**

| | 3ʳᵈ Trimester Visit 1 (n = 59) | 4-6 Weeks PP Visit 2 (n = 66) | 4 Months PP Visit 3 (n = 60) | 8 Months PP Visit 4 (n = 60) | 12 Months PP Visit 5 (n = 56) |
|---|---|---|---|---|---|
| **Actigraphy-measured sleep** | | | | | |
| Time in Bed (hours) | 8.44 (0.91) | 8.66 (1.19) | 8.09 (0.96) | 8.06 (0.98) | 7.97 (0.77) |
| Total Sleep Time (hours) | 7.27 (0.89) | 6.87 (0.89) | 6.86 (0.86) | 6.90 (0.82) | 6.78 (0.83) |
| Wake After Sleep Onset (min) | 70.25 (26.26) | 107.42 (37.14) | 73.80 (22.98) | 68.56 (23.13) | 71.76 (29.61) |
| Sleep Latency (min) | 9.17 (6.91) | 8.60 (6.28) | 8.56 (7.57) | 8.48 (5.66) | 8.45 (5.05) |
| Sleep Efficiency (%) | 86.17 (5.13) | 79.88 (5.50) | 85.00 (4.19) | 85.85 (3.94) | 85.13 (6.04) |
| **Subjective sleep (PQSI)** | | | | | |
| Time in Bed (hours) | 8.49 (1.32) | 8.52 (1.93) | 7.85 (1.15) | 7.72 (1.39) | 8.15 (1.55) |
| Total Sleep Time (hours) | 6.70 (1.18) | 5.86 (1.37) | 6.29 (1.21) | 6.27 (1.24) | 6.83 (1.34) |
| Sleep Latency (min) | 21.59 (18.60) | 14.17 (11.78) | 16.21 (11.49) | 19.00 (13.20) | 22.13 (26.21) |
| Sleep Efficiency (%) | 79.33 (11.73) | 69.44 (14.46) | 80.05 (12.54) | 81.67 (14.37) | 84.18 (12.46) |
| PSQI Global Score* | 6.95 (3.00) | 7.73 (2.72) | 6.62 (2.86) | 6.62 (2.88) | 5.93 (3.18) |

* PSQI Global Score ranges from 0–21, where a higher score indicates worse sleep and > 5 used as a cutoff for poor sleep quality.

There was a significant linear (b = −4.22, 95% CI [−7.11, −1.31], $p$ = .004) and quadratic (b = 0.93, 95% CI [0.45, 1.40], $p$ < .001) effect of time, and a significant difference based on method of data collection (b = −5.63, 95% CI [−7.18, −4.08], $p$ < .001) such that the actigraphic measure of sleep consistently estimated a greater sleep efficiency than self-report measures, which is in line with the finding that actigraphy captured a shorter time to fall asleep (latency) and longer sleeping times overall compared with the self-report measures. There was not a significant interaction between time (linear or quadratic) and method of data collection ($p$s > .16; see Fig 2C).

## Sleep latency

Sleep latency was consistently lower by actigraphy compared to self-report, ranging from 8.45–9.17 minutes by actigraphy and 14.17–22.13 minutes by self-report (see Table 2; see Fig 1D). Actigraphic and self-reported sleep latency were not reliably correlated across any study visits: 3ʳᵈ trimester ($r$ = .17, $p$ = .19), 4−6 weeks postpartum ($r$ = .14, $p$ = .27), 4 months postpartum ($r$ = −.02, $p$ = .87), 8 months postpartum ($r$ = .25, $p$ = .06), nor 12 months postpartum ($r$ = −.06, $p$ = .63).

Overall, there was a significant linear (b = −4.42, 95% CI [−7.88, −0.96], $p$ = .013) and quadratic effect of time (b = 0.79, 95% CI [0.23, 1.35], $p$ = .006) related to sleep latency, and a significant difference based on method of data collection (b = 9.82, 95% CI [8.02, 11.62], $p$ < .001) such that the actigraphic measure of sleep captured less time awake before the sleep period began than was self-reported by the participants. There was also a significant interaction between time (linear) and method (b = −8.25, 95% CI [−14.82, −1.67], $p$ = .015), as well as time (quadratic) and method (b = 1.50, 95% CI [0.42, 2.58], $p$ = .007; see Fig 2D); an analysis of simple slopes indicated that this difference was significant at each visit: 3ʳᵈ trimester of pregnancy (b = 12.42, $p$ < .001), 4–6 weeks postpartum (b = 5.57, $p$ < .001), 4 months postpartum (b = 7.65, $p$ < .001), 8 months postpartum (b = 10.53, $p$ < .001), and 12 months postpartum (b = 13.68, $p$ < .001).

## Wake after sleep onset

Wake after sleep onset (actigraphy only), ranged from 69.56–107.42 minutes (see Table 2; see Fig 1E). There was a significant linear (b = 15.49, 95% CI [3.50, 27.45], $p$ = .011) and quadratic (b = −3.22, 95% CI [−5.18, −1.26], $p$ = .001) effect of time for wake after sleep onset.

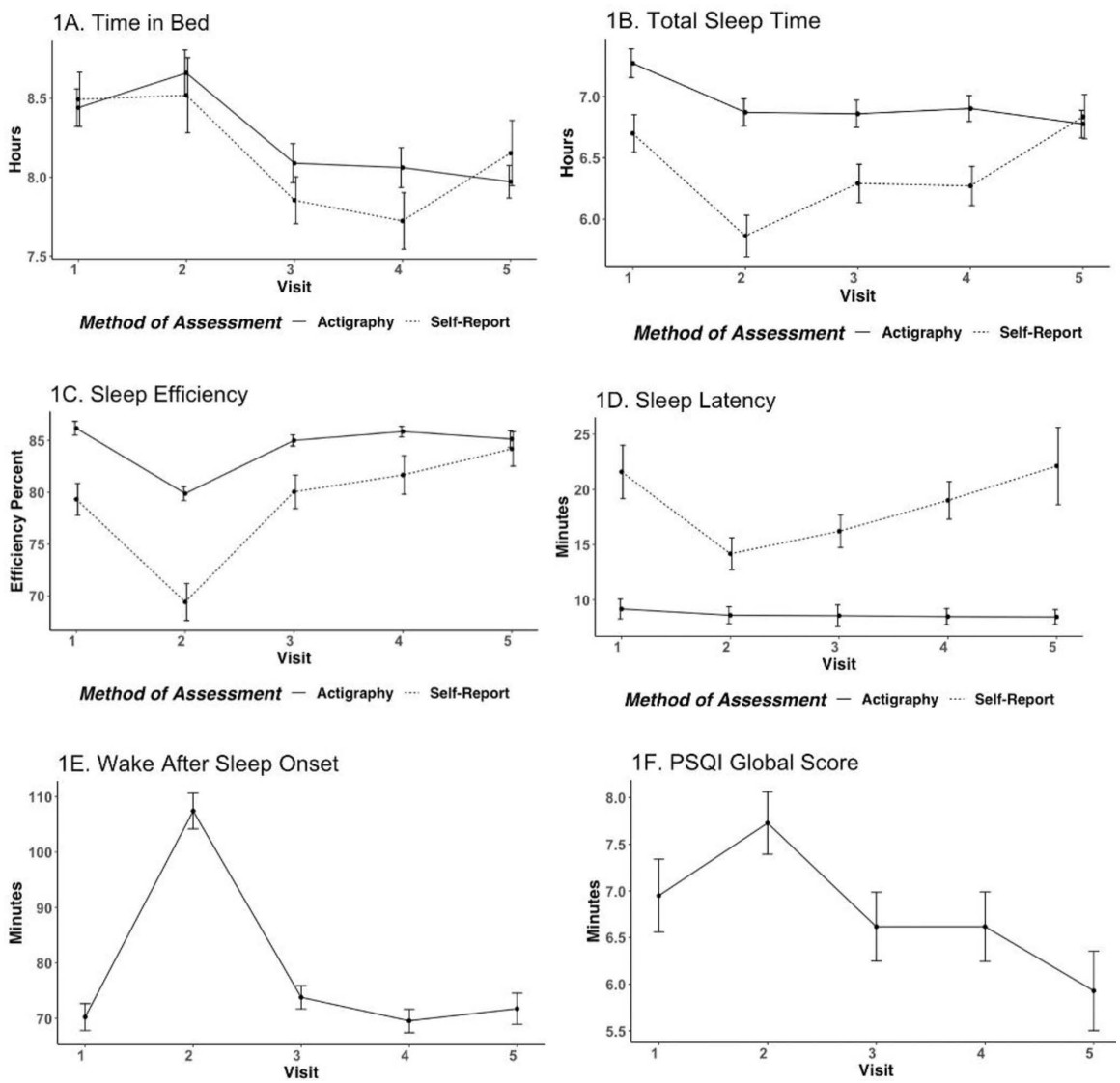

**Fig 1. A-F. Mean (SE) values of time in bed (TIB), total sleep time (TST), sleep efficiency (SE), sleep latency (SL), wake after sleep onset (WASO), and PSQI score over visits.**

### PSQI global score

The PSQI Global Score (self-report only), ranged from 5.93–7.73 (see Table 2; see Fig 1F). There was a significant quadratic (b = −0.17, 95% CI [−0.31, −0.03], p = .015), but not linear (p = .10) effect of time related to the PSQI Global Score.

### Discussion

Among 74 women sampled five times from their 3rd trimester of pregnancy to 12 months postpartum, we identified differences by sleep measurement method for total sleep time, sleep efficiency, and sleep latency. There were no significant differences for time in bed regardless of whether it was assessed via self-report (i.e., PSQI) or actigraphy. Broadly, actigraphy tended to reflect longer total sleep times, shorter time to fall asleep (sleep latency), and greater sleep efficiency

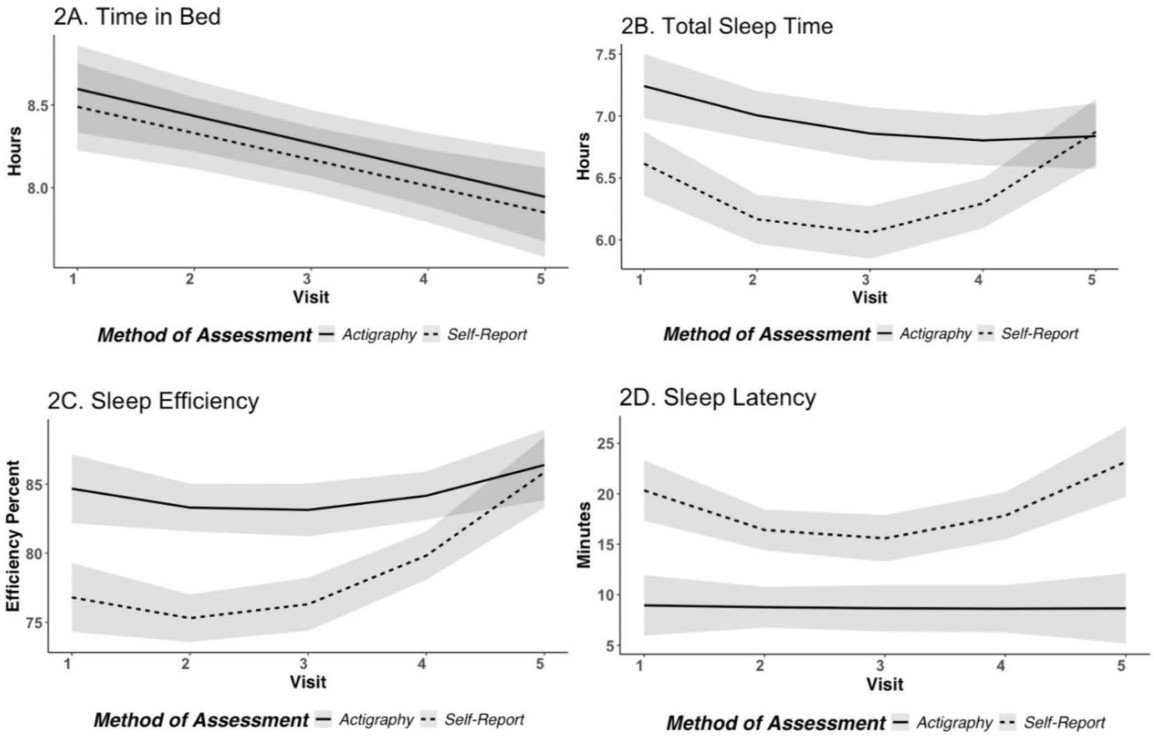

**Fig 2. A-D. Comparison of sleep assessment methods for TIB, TST, SE, and SL.** This graph shows model-based predicted marginal means generated from the ggpredict() function in the *ggeffects* R package (42). Shaded regions represent the 95% confidence intervals for the predicted marginal means.

compared to self-reported measures. Understanding how different measurement modalities may affect sleep data collected during pregnancy and postpartum is important for researchers in this area.

Overall, these findings align with other studies of sleep through pregnancy or postpartum where there are decreases in total sleep time and sleep efficiency and increases in wake after sleep onset during the postpartum period (compared to pregnancy), particularly in the first 2–3 months postpartum [43,44]. Our findings indicate that self-reported sleep duration may be underestimated compared to actigraphy in the perinatal population, however the two methods were low to moderately correlated. Others have also shown a moderate correlation between actigraphy-assessed sleep and self-reported sleep in non-pregnant samples [45,46], but these findings are not uniform across the literature and agreement between measurement modalities may also depend on psychological factors such as depressive symptoms, hostility, and perceptions of overall health [47]. Prior research comparing actigraphic and self-reported total sleep time in mid-pregnancy identified that self-reported total sleep time was overestimated compared to actigraphic total sleep time [21]; however, the authors also identified variability in the direction of this effect as 23% underestimated their sleep time by more than an hour while 39% of the sample overestimated their sleep time compared to actigraphy. Variability in self-report measures of sleep is worth considering when implementing sleep diaries; however, it is also possible that there is measurement error from actigraphy that may be contributing to these differences, and that some of this measurement error could be unique to the perinatal period.

While sleep latency and sleep efficiency were consistently estimated differently based on the mode of measurement across the study period, estimates of total sleep time converged by 12 months postpartum. This may reflect less disrupted sleep by 12 months postpartum as infants sleep more throughout the night or may reflect a more general recovery of sleep as one may have developed better strategies to fall back asleep by this point. The differences in sleep latency have

similarly been found in other studies, across various populations, and among pregnant women [48–50]. Individuals often overestimate the amount of time it takes them to fall asleep and actigraphy depends on movements for its measurements and will inherently underestimate due to minimal movements when attempting to fall asleep. Assessing and treating sleep difficulties in pregnancy and postpartum is a priority for health of both the mother and developing fetus/infant. Indeed, our results highlight the clinical challenge of using actigraphy in samples that experience substantial time awake during the night.

There are several limitations to our interpretation of these data. First, we did not use polysomnography, which is the gold standard for sleep measurement but not practical for measuring sleep in home; instead, we focused on actigraphy as a well-validated, less burdensome, and more commonly used method of sleep assessment in population studies. The self-reported measure of sleep (PSQI) asks participants to reflect on their sleep over the past four weeks whereas actigraphy was only measured for a week. We do not know how much of differences in the two methods may be from error related to retrospective reporting or error that is inherent when self-reporting sleep. We also were not able to capture sleep disorders that may have been present for some of the women in this sample (e.g., restless leg syndrome, sleep disordered breathing). Lastly, while our sample was larger than some previous studies, it remains a relatively small number of participants and may not be generalizable to other pregnant and postpartum women. For example, in our sample, most women had a prior birth and the variability in the age of that child in their home may affect the sleep experiences reported here.

These limitations are well-balanced by several critical strengths. This study examined the congruence between self-report and actigraphy-assessed sleep in repeated measurements capturing sleep-related changes in the transition to parenthood, which is a sensitive period for women's health. The only previous examination of perinatal concordance in sleep measurement is limited to pregnancy and did not include the PSQI [30]; thus, our results nicely complement and extend the available literature.

Given the elevated mental health risks of the postpartum period and the well-established links between sleep and mental health risk, many researchers are measuring sleep and mental health symptomology in the postpartum period; our results can inform the ways that choosing to measure sleep via self-report using the PSQI or actigraphy alone may influence various metrics of sleep health. In summary, this study identified that actigraphy-assessed perinatal sleep may indicate longer total sleep times, shorter sleep latency, and greater sleep efficiency compared to self-reported sleep. Sleep latency and efficiency were consistently different across the mode of measurement throughout the entire study period whereas total sleep time was only different by mode of measurement in the first eight months following the transition to parenthood. Using longitudinal assessment of sleep health and knowing the implications of different methods for measurement is critical to advance research on maternal and infant health.

## Supporting information

**S1 Data. Full data for reproducing reported findings.**
(CSV)

## Acknowledgments

We appreciate the contributions of our Clinical Research Assistants and students to data collection. We also thank our study participants and staff at the Ohio State University Wexner Medical Center Prenatal Clinic.

## Author contributions

**Conceptualization:** Lisa M. Christian.

**Formal analysis:** Ryan L. Brown.

**Funding acquisition:** Lisa M. Christian.

**Supervision:** Lisa M. Christian.

**Visualization:** Ryan L. Brown.

**Writing – original draft:** Ryan L. Brown, Chloe M. Beverly Hery.

**Writing – review & editing:** Ryan L. Brown, Aric A. Prather, Lisa M. Christian.

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
