## [Decision Letter · Decision Letter 0]

15 Dec 2024

PONE-D-24-36655Comparison of longitudinal PSQI and actigraphy-assessed sleep from pregnancy to postpartumPLOS ONE

Dear Dr. Brown,

Thank you for submitting your manuscript to PLOS ONE. After careful consideration, we feel that it has merit but does not fully meet PLOS ONE’s publication criteria as it currently stands. Therefore, we invite you to submit a revised version of the manuscript that addresses the points raised during the review process.

We look forward to receiving your revised manuscript.

Kind regards,

Sascha Köpke

Academic Editor

PLOS ONE

“This study was supported by the Eunice Kennedy Shriver National Institute of Child Health and Human Development (NICHD) [grant numbers HD061644 and HD067670 to L.M.C.]. The project described was also supported by Award Number UM1TR004548 from the National Center for Advancing Translational Sciences. The content is solely the responsibility of the authors and does not necessarily represent the official views of the National Center for Advancing Translational Sciences or the National Institutes of Health.”

3. We note that there is identifying data in the Supporting Information file <datafile_PLOSone.csv>. Due to the inclusion of these potentially identifying data, we have removed this file from your file inventory. Prior to sharing human research participant data, authors should consult with an ethics committee to ensure data are shared in accordance with participant consent and all applicable local laws.

-Location data

Please remove or anonymize all personal, ensure that the data shared are in accordance with participant consent, and re-upload a fully anonymized data set. Please note that spreadsheet columns with personal information must be removed and not hidden as all hidden columns will appear in the published file.

Reviewers' comments:

Reviewer's Responses to Questions

**Comments to the Author**

1. Is the manuscript technically sound, and do the data support the conclusions?

Reviewer #1: Yes

Reviewer #2: Yes

2. Has the statistical analysis been performed appropriately and rigorously? 

Reviewer #1: Yes

Reviewer #2: Yes

3. Have the authors made all data underlying the findings in their manuscript fully available?

Reviewer #1: Yes

Reviewer #2: Yes

4. Is the manuscript presented in an intelligible fashion and written in standard English?

Reviewer #1: Yes

Reviewer #2: Yes

5. Review Comments to the Author

Reviewer #1: I commend the effort to explore this area of research, as it represents an innovative and valuable contribution to the field. However, the paper has minor limitations that need to be addressed to strengthen.

Reviewer #2: Overall, this is a really good paper. It is an interesting and important topic. The methods and the analysis seem sound. It’s very clearly and concisely written and the conclusions seem well-founded. I have some comments and suggestions for the authors below, but would hope this will be published with only relative minor revisions.

Broader points

It would be good to provide a brief review of some literature on the concordance between actigraphy and self-report data in other populations (most obviously healthy working age adults). Though some of the parameters converge at visit-5, there are lots of studies out there finding lack of concordance outside the perinatal period.

It would be nice to have some further consideration of the correlational data – they are not really presaged in the introduction/ method or discussed greatly in the discussion. I’ve found in the past that non-specialist readers struggle to distinguish between sleep measures that concord in their measurement of individual differences and those that concord in their absolute parameter estimates.

There is no mention of sleep diaries being used in the protocol. This would typically be best practice for cleaning actigraphy data. I assume it wasn’t included here to avoid participant burden, but should be considered as a limitation.

Introduction

The authors may want to justify the focus on US studies – i.e. are they highlighting potential cultural determinants of perinatal sleep?

On reference to PSG: I’d also question whether polysomnography would be a good way to measure habitual sleep, which is the target here (e.g. short number of nights, inconsistent environment).

Method

Do the authors have data on the number of women approached?

3 nights of actigraphy is quite a low bar for reliability – Would there be merit in the authors running sensitivity analysis on the number of nights collected?

The data are obviously now quite old, can the authors confirm that the data are not published elsewhere, and/ or cite related publications?

Could the authors be clearer on the cleaning process – what were the criteria for “decreased activity levels” for instance? The worry is that you might have inter-rater reliability based on training coders in a process that is not valid.

Results

The time points have some variation in the gaps between them – would this impact the analysis?

Is there not a way to extract WASO from the PSQI data?

Is it possible to be more explicit on the treatment of multiple comparisons?

Discussion

Is there scope to discuss sleep latency in more detail (with reference to the literature) - here the discordance seems particularly striking and therefore may be more methodological than time period specific. I have some thoughts as to why, but would imagine the authors do to, that are probably better informed (so won't be prescriptive).

Would it help to discuss the sources of the discordance more? I felt the main tone of the discussion was that this was subjective innaccuracy, but is there a role for actigraphic innacuracy in this population (that wakes a lot and may regularly remain still while settling a baby)?

6. PLOS authors have the option to publish the peer review history of their article (what does this mean? ). If published, this will include your full peer review and any attached files.

**Do you want your identity to be public for this peer review?** For information about this choice, including consent withdrawal, please see our Privacy Policy .

Reviewer #1: **Yes: ** Esuyawkal Mislu

Reviewer #2: No

---

## [Author Response · Author response to Decision Letter 1]

20 Feb 2025

Reviewer #1: I commend the effort to explore this area of research, as it represents an innovative and valuable contribution to the field. However, the paper has minor limitations that need to be addressed to strengthen.

We appreciate Reviewer 1’s feedback, as well as their time and effort to improve this paper. We have updated the manuscript in accordance with their comments below.

Abstract

1. The abstract has no introduction

We added to the beginning of the abstract to provide an overarching introduction to the purpose of the study.

From Abstract, Page 2:

“Despite the importance of sleep for perinatal health, there is limited research examining whether different measurement modalities may yield inconsistent data from pregnancy through postpartum. We aimed to: 1) describe how sleep patterns change across pregnancy and postpartum using self-report PSQI and actigraphy measures and 2) determine the level of correspondence between these two measurement modalities.”

2. The statistical methods used to analyze differences between actigraphy and PSQI measures were not explained.

We have added the primary statistical method to the abstract.

From Abstract, Page 2:

“Using generalized linear mixed modeling, we examined if there were differences by sleep measurement (actigraphy vs. PSQI) for the overlapping sleep outcomes: total sleep time, time in bed, sleep latency, and sleep efficiency.”

3. The cause of this difference was not explained

We now describe potential reasons for discrepancies across the measurement modalities in the conclusions section of the abstract.

From Abstract, Page 2:

“Actigraphy-assessed sleep may reflect longer total sleep times, shorter sleep latency, and greater sleep efficiency compared to self-reported sleep among pregnant and postpartum women. This may be due to measurement error in actigraphy or recall bias when completing self-reported sleep measures.”

4. The recommendation is not in line with the objective

We have adjusted this recommendation to be more focused on the importance of knowledge of how measurement decisions may influence estimates of sleep-related parameters in the perinatal period.

From Abstract, Pages 2-3:

“Actigraphy-assessed sleep may reflect longer total sleep times, shorter sleep latency, and greater sleep efficiency compared to self-reported sleep among pregnant and postpartum women. This may be due to measurement error in actigraphy or recall bias when completing self-reported sleep measures. These factors should be taken into consideration both at the time of study design and when comparing results from different studies to facilitate the highest quality research and clinical decision-making in this population.”

Introduction

1. Line 65, 66 , reference 2, 4 is an old reference

These two references have now been replaced with more recent ones: (Lu et al., 2021), (Sedov et al., 2018) and (Sobol et al., 2024).

2. The paragraph “Six studies in the United States have examined changes in parental sleep from pregnancy 93 to postpartum using actigraphy [24-29], of which four had small sample sizes of ≤30 participants 94 [26-29]. Additionally, none of these studies have examined sleep after 16 weeks postpartum, 95 when many sleep complaints do not begin to occur until the third trimester (after 28 weeks) and 96 few report sleep metrics beyond total sleep time and sleep efficiency.” is not necessary. It is better to for the introduction focus on the main objective.

Thank you for this comment as it indicated to us that we had not been clear enough about why this information was included and indeed part of the main objective (and strength) of this study. We amended this paragraph to more explicitly indicate that the lack of studies capturing sleep in the perinatal period comprehensively is a challenge for researchers, and that our study addresses this by continuing through the first year postpartum.

From Introduction, Page 5:

“When considering actigraphy-assessed sleep throughout the perinatal period, there are several limitations to the available literature such as small samples and a lack of postpartum assessments. Six studies in the United States have examined changes in parental sleep from pregnancy to postpartum using actigraphy [25-30], of which four had small sample sizes of ≤30 participants [27-30]. Additionally, none of these studies have examined sleep after 16 weeks postpartum, when many sleep complaints do not begin to occur until the third trimester (after 28 weeks) and few report sleep metrics beyond total sleep time and sleep efficiency. In the current study, we report actigraphy-assessed sleep from third trimester to one year postpartum to address this gap in postpartum measurement.”

3. The result of previous studies was not present quantitatively, and need explanation on the type of difference.

The two key studies (Herring 2014 and Okun 2021) are now presented in the introduction with the quantitative differences described.

From Introduction, Pages 5-6:

“Over 60% of participants self-reported sleep duration estimates that differed by over an hour compared to actigraphy (21), actigraphy-assessed wake after sleep onset was 3.5 times greater than self-reported (31), and self-reported sleep efficiency was much higher than actigraphy-assessed (31)”

4. The limitation of measurement error from actigraphy and their cause was not explained as a comparison with the gold standard or patient report.

The introduction presents other studies that have compared actigraphy and self-report to help focus the objective of our current study. Actigraphy has been consistently shown to have high concordance with polysomnography (the gold standard); however, our study did not include a comparison with PSG (particularly given in home measurement) and therefore did not describe findings of previous studies for PSG vs actigraphy. The authors are happy to add more about PSG if the reviewer feels necessary.

Method

Well written

Appropriate statistical analysis was done, with better interpretation.

However, the explanation related to PSQI (Line 141-149) is too broad, write only the main ideas.

We are glad that the methods section read well overall. We have removed the extraneous information about the PSQI subscales and subscale scoring.

Reviewer #2: Overall, this is a really good paper. It is an interesting and important topic. The methods and the analysis seem sound. It’s very clearly and concisely written and the conclusions seem well-founded. I have some comments and suggestions for the authors below, but would hope this will be published with only relative minor revisions.

Thank you for your kind feedback on our paper, particularly the writing and conclusions. We appreciate you taking the time to provide the feedback below to strengthen it for publication.

Broader points

It would be good to provide a brief review of some literature on the concordance between actigraphy and self-report data in other populations (most obviously healthy working age adults). Though some of the parameters converge at visit-5, there are lots of studies out there finding lack of concordance outside the perinatal period.

We have added to our discussion to situate the results within the broader context of sleep research and psychosocial predictors of differences in non-pregnant samples.

From Discussion, Page 19:

“Others have also shown a moderate correlation between actigraphy-assessed sleep and self-reported sleep in non-pregnant samples [45, 46], but these findings are not uniform across the literature and agreement between measurement modalities may also depend on psychological factors such as depressive symptoms, hostility, and perceptions of overall health [47].”

It would be nice to have some further consideration of the correlational data – they are not really presaged in the introduction/ method or discussed greatly in the discussion. I’ve found in the past that non-specialist readers struggle to distinguish between sleep measures that concord in their measurement of individual differences and those that concord in their absolute parameter estimates.

Correlations were conducted to present the initial concordance between each modality at each time point, cross-sectionally, without accounting for the time or individual effects. Correlations can also support meta-analytic investigations in the future. The main outcomes are our GLMM models which can account for repeated measurements from the same person; therefore, these more reliable models were the focus of our results and discussion.

There is no mention of sleep diaries being used in the protocol. This would typically be best practice for cleaning actigraphy data. I assume it wasn’t included here to avoid participant burden, but should be considered as a limitation.

Sleep diaries on their own provide researchers with a good understanding of how individuals perceive their sleep and their general sleep duration. However, sleep diaries are notorious for low adherence and recall bias, and are often not used to confirm sleep periods in actigraphy. We utilized a validated actigraphy scoring method, which has been published and used in multiple other studies. We added to the manuscript to indicate that sleep diaries were collected but rarely used (only when sleep periods were difficult to determine).

From Materials and methods, Page 9:

“Sleep diary data was used to help suggest, but not confirm, sleep periods as needed.”

Introduction

The authors may want to justify the focus on US studies – i.e. are they highlighting potential cultural determinants of perinatal sleep?

Thank you for bringing up this point. There are likely cultural differences surrounding sleep and the perinatal period (e.g., differences in maternity leave), however since our population was US based, the introduction also focuses on the US population.

On reference to PSG: I’d also question whether polysomnography would be a good way to measure habitual sleep, which is the target here (e.g. short number of nights, inconsistent environment).

We agree that PSG would not have been a good choice for this population due to the reasons you mention above.

Method

Do the authors have data on the number of women approached?

We do not have information on the number of women approached. Recruitment efforts used advertisements (among other method) which makes it difficult to quantify how many women were reached.

3 nights of actigraphy is quite a low bar for reliability – Would there be merit in the authors running sensitivity analysis on the number of nights collected?

Participants were requested to wear the sleep watch for 7 days. If participants did not wear the watch each night, only participants with at least 3 days of valid data were included to ensure habitual sleep was captured. At least 3 valid nights is standard in actigraphy studies. However, across all five study visits, there are only 23 cases (~8% of data points) of individuals with 3-4 nights of actigraphy data recorded. The average participant had between 6.3 and 6.6 valid days of actigraphy across the five study visits. To be sure that the threshold of the number of nights collected was not biasing our primary results, we ran sensitivity analyses including only those with 5 or more valid nights of actigraphy and can confirm that the results are unchanged.

From Materials and methods, Page 7:

“Across all five study visits, there were only 23 cases (~8% of data points) of individuals with only three to four nights of actigraphy data recorded.”

From Materials and methods, Page 10:

“In post-hoc sensitivity analyses, we confirmed that the presented results were unchanged when restricting the sample to only those who had five or more valid nights of actigraphy.”

From Results, Page 11:

“On average these participants had between 6.3 and 6.6 valid days of actigraphy.”

The data are obviously now quite old, can the authors confirm that the data are not published elsewhere, and/ or cite related publications?

We can confirm that these data are not published anywhere else.

Could the authors be clearer on the cleaning process – what were the criteria for “decreased activity levels” for instance? The worry is that you might have inter-rater reliability based on training coders in a process that is not valid.

Actigraphy scorers used a validated procedure (see: Marino M, Li Y, Rueschman MN, et al. Measuring sleep: Accuracy, sensitivity, and specificity of wrist actigraphy compared to polysomnography. Sleep 2013;36:1747–55) and scoring methods, including cleaning, which has been conducted across multiple studies with various populations (a few selected publications: https://pmc.ncbi.nlm.nih.gov/articles/PMC10524712/;
https://pmc.ncbi.nlm.nih.gov/articles/PMC9019820/;
https://pmc.ncbi.nlm.nih.gov/articles/PMC5020369/).

Decreasing activity levels refers to the scorers considering how the activity intensity decreases surrounding periods of suspected sleep within the context of the entire participant’s actigraphy profile. Scorers visually reviewed the entire sleep recording before manually adjusting any sleep periods to consider the activity intensity level of the participant to ensure periods of low movement, but not sleep (e.g. reading in bed, watching TV) were not included.

We have added to the information in the manuscript in case other readers may have similar questions.

From Materials and methods, Pages 8-9:

“Sleep actigraphy data files for each participant were scored in Actiware using the validated manufacturer’s standard algorithm at medium sensitivity [34]. Two independent scorers determined the validity of each recording based on standard sleep criteria, adapted from other actigraphy studies [36-38].”

“For example, decreasing activity levels refers to the scorers considering how the activity intensity decreases surrounding periods of suspected sleep within the context of the entire participant’s actigraphy profile. Scorers visually reviewed the entire sleep recording before manually adjusting any sleep periods to consider the activity intensity level of the participant to ensure periods of low movement, but not sleep (e.g. reading in bed, watching TV) were not included.”

Results

The time points have some variation in the gaps between them – would this impact the analysis?

While we understand the reviewer’s concern, the uneven intervals here were designed purposefully to best capture changes at meaningful time points. Thus, closer intervals were used near the time of delivery, at which time there is the greatest change in the women’s lives in relation to recovering from childbirth and also caring for a newborn. Our primary questions were focused on within time point comparisons between the different measurement modalities rather than if the measurement modalities aligned across the full study period.

Is there not a way to extract WASO from the PSQI data?

No, unfortunately there is not a way to extract WASO from the PSQI data that can be compared to actigraphy data. There are individual items that ask about waking after falling asleep or there is the subscale component 5 on sleep disturbance, but these do not give an estimated amount of WASO in minutes which could be compared to actigraphy WASO.

Is it possible to be more explicit on the treatment of multiple comparisons?

The current study did not adjust for multiple comparisons. However, all analyses were planned a priori. Moreover, analyses focus on between modality comparisons as each given timepoint, rather than comparisons across all measurement timepoints, thus reducing the total number of tests. We have not added the manuscript due to length, but would be happy to add this as a limitation if the reviewer/editor would prefer this addition.

Discussion

Is there scope to discuss sleep latency in more detail (with reference to the literature) - here the discordance seems particularly striking and therefore may be more methodological than time period specific. I have some thoughts as to why, but would imagine the authors do to, that are probably better informed (so won't be prescriptive).

Thank you for pointing this out – we have added more discussion about

---

## [Decision Letter · Decision Letter 1]

9 Apr 2025

Comparison of longitudinal PSQI and actigraphy-assessed sleep from pregnancy to postpartum

PONE-D-24-36655R1

Dear Dr. Christian,

We’re pleased to inform you that your manuscript has been judged scientifically suitable for publication and will be formally accepted for publication once it meets all outstanding technical requirements.

Kind regards,

Sascha Köpke

Academic Editor

PLOS ONE

Additional Editor Comments (optional):

Reviewers' comments:

Reviewer's Responses to Questions

**Comments to the Author**

1. If the authors have adequately addressed your comments raised in a previous round of review and you feel that this manuscript is now acceptable for publication, you may indicate that here to bypass the “Comments to the Author” section, enter your conflict of interest statement in the “Confidential to Editor” section, and submit your "Accept" recommendation.

Reviewer #1: All comments have been addressed

2. Is the manuscript technically sound, and do the data support the conclusions?

Reviewer #1: Yes

3. Has the statistical analysis been performed appropriately and rigorously? 

Reviewer #1: Yes

4. Have the authors made all data underlying the findings in their manuscript fully available?

Reviewer #1: Yes

5. Is the manuscript presented in an intelligible fashion and written in standard English?

Reviewer #1: Yes

6. Review Comments to the Author

Reviewer #1: (No Response)

7. PLOS authors have the option to publish the peer review history of their article (what does this mean? ). If published, this will include your full peer review and any attached files.

**Do you want your identity to be public for this peer review?** For information about this choice, including consent withdrawal, please see our Privacy Policy .

Reviewer #1: **Yes: ** Esuyawkal Mislu

---

## [Editor Report · Acceptance letter]

PONE-D-24-36655R1

PLOS ONE

Dear Dr. Christian,

I'm pleased to inform you that your manuscript has been deemed suitable for publication in PLOS ONE. Congratulations! Your manuscript is now being handed over to our production team.

Kind regards,

on behalf of

Professor Sascha Köpke

Academic Editor

PLOS ONE